# Perspectives of adolescents and young people on Digital Health Interventions and their impact on health knowledge

Paul Macharia[1,2,3,4,5]*, Maiya G. Block Ngaybe[6], Priyanka Ravi[7], Hellen Moraa[2], Wadana Hamzazai[8], Boyani Moikobu[5], Cyrus Mugo[2‡], Christine Ngaruiya[9‡]

**1** Strathmore University, Nairobi, Kenya, **2** University of Nairobi, Nairobi, Kenya, **3** Kenyatta National Hospital, Nairobi, Kenya, **4** Stanford University, Center for Innovation in Global Health, Palo Alto, United States of America, **5** Consulting in Health Informatics, Nairobi, Kenya, **6** School of Landscape Architecture, College of Architectural Planning and Landscape Architecture, University of Arizona, Tucson, Arizona, United States of America, **7** Department of Education, Health, and Behavior Studies, College of Education and Human Development, University of North Dakota, Grand Forks, North Dakota, United States of America, **8** Department of Biostatistics and Epidemiology, Mel and Enid Zuckerman College of Public Health, University of Arizona, Tucson, Arizona, United States of America, **9** Department of Emergency Medicine, Stanford School of Medicine, Palo Alto, California, United States of America

‡ Senior authors.
* paulmachariah@gmail.com

## Abstract

Adolescents and young people (AYP) in Kenya face unique health challenges, but there is a lack of research on their health knowledge and awareness needs. Digital health interventions (DHIs) could help address these gaps. Understanding AYP's current knowledge will inform the development of effective, adolescent-centered interventions. Grounded in the inter-agency framework from WHO, UNICEF, UNFPA, and UNESCO for youth-centered DHIs, this study assessed AYP's health knowledge and awareness in three regions of Kenya. The study focused specifically on HIV, intimate partner violence (IPV), substance use, mental health, sexual health, and nutrition. Additionally, it evaluated AYP's preferences for and use of DHIs. This qualitative study used focus group discussions to assess health knowledge and awareness among AYP 19–24 years of age in Kibra (urban), Kikuyu (peri-urban), and Nachu (rural) Kenya. Participants were purposively selected. Data analysis involved independent coding using MAXQDA and thematic analysis to identify key themes. Seventy participants were included in the study with almost similar gender distribution of 36 female and 34 male participants. Young people in Kenya utilize online resources and apps for health information. Despite a general awareness of health issues, knowledge gaps exist concerning HIV prevention, stigma, and treatment. Participants also linked substance use with mental health problems. Cost of internet use and misinformation were barriers to using DHIs. The participants expressed a preference for future DHIs to enable interaction with peers and experts, include referral services, and prioritize privacy. Our study highlights that a targeted health-related app

**Data availability statement:** A deidentified data set may be made available upon request from author PM or MBN. Due to the sensitive nature of the topics of conversation, it is better to keep the dataset private to protect the identities of the participants. Dr. Danny Nyatuka, Lecturer, School of Computing and Engineering Sciences may also be contacted for access to data if need be at dnyatuka@strathmore.edu. The data are being stored securely online in a shared secure folder which remains accessible to authors on this study should anyone reach out to request access to data.

**Funding:** This work was supported by the National Institutes of Health, Fogarty International Center/Global Health Equity Scholars Fellowship (D43TW010540 for MBN and PM). The funders had no role in study design, data collection and analysis, decision to publish, or preparation of the manuscript.

**Competing interests:** The authors have declared that no competing interests exist.

could benefit many adolescents across Kenya. The participatory design of our study was a notable strength. However, future studies could benefit from a less structured interview guide, allowing for deeper understanding of less common health issues. This research will inform the development of a health-focused DHI for Kenyan AYPs, utilizing a user-centered design approach.

## Background

Adolescents and young people are among the growing number of people living with human immunodeficiency virus (HIV) globally [1]. In 2020, it was estimated that about 410,000 persons aged between 10 – 24 years were newly infected with HIV [2]. The World Health Organization (WHO) in 2021 estimated that about 1.7 million adolescents and young people were living with HIV with 90% in WHO African region [3]. Despite advancement in HIV care, adolescents continue to face social stigma and barriers to care, resulting in poor treatment outcomes [4].

Efforts by the Joint United Nations Program on HIV/AIDS (UNAIDS) supported by governments to improve HIV awareness among adolescents and young people is bearing fruit as a number of research findings show [5]. In one study in Nigeria, research findings showed that adolescents had high knowledge and awareness on HIV. This resulted in low-risk behavior [6]. In one Ugandan study, it was observed that exposure to information and awareness on HIV potentially increased testing for HIV among adolescents [7]. In another Ugandan study, significant improvement on knowledge and awareness on HIV among adolescents was noted. The use of the internet to access much needed information seems to have had a positive impact [8].

Despite the great success, a gap still exists in HIV knowledge, medication-taking literacy, mental health support and treatment adherence [9]. Adolescents and young people also need safe spaces to interact and share information about HIV and reproductive health, in addition to developing the skills necessary to interact with healthcare providers, navigate the health system, and critically appraise health-related information from diverse sources [10].

The International Telecommunication Union (ITU) estimated that worldwide, in 2020, 71% of young people aged 15 – 19 years accessed the internet with about 40% of these young users being from Africa [11]. Most of the internet use by adolescents is on social media platforms and applications [12]. Appropriate use of the internet facilitates communication, socialization and entertainment among adolescents [13]. Social media has also been used to improve knowledge on HIV among adolescents in resource-limited settings [14].

In one of such settings, an interaction forum was used by people living with HIV and their loved ones to share information and support on topics such as diagnosis, treatment, daily living and community resources [15]. In another setting that attempted to use online forums to combat stigma, findings showed that online forums to address stigma could improve efficacy in the HIV prevention and care continuum [16].

There exists a growing demand by adolescents and young people to easily interact, peer-learn and share experiences on HIV prevention, care, treatment and

support. Digital health interventions (DHIs) offer great potential [17]. Social media platforms have been used to promote HIV testing and provide opportunities to reach unreached populations in a relevant, effective and safe manner [18–20]. In one Kenyan study, results showed that social media use to offer health interventions to adolescents was promising and should be explored as a platform to deliver youth-targeted programs [21].

Despite the great success and willingness to use social media apps to access HIV information, fear of inadvertent disclosure of their HIV status during interactions exists [22]. A number of adolescents are less inclined to use current social media apps to interact on HIV related discussions fearing their online activities could be viewed by others [23]. To note is that, despite potential benefits of the social media interaction, these are outweighed by privacy and confidentiality concerns [24].

Research has also demonstrated that a user-centric approach to the design and development of DHIs targeting adolescents have potential to scale-up [25]. Online social media platforms have empowered adolescents living with HIV by making them feel connected, safe and able to interact with their peers [26]. Web-based delivery of DHIs could make it possible for adolescents and young people to access information, group members privately, when convenient, and without travel [14]. The nature of digital health also allows adolescents to build friendships beyond their geographic area and interact with their peers in real time [27]. This study used a human-centered design (HCD) approach employing qualitative methods. The HCD approach is a collaborative process involving participatory design, ethnography and contextual design [28]. The objective of this study was to understand the use, preferences and perspectives of adolescents and young people regarding digital health interventions and the potential impact of these interventions on health knowledge.

## Methodology

### Study recruitment

A convenient sample of participants was enrolled from the three study settings. This approach facilitated the recruitment of sufficient participants within the required timeframe to achieve thematic saturation, ensuring a comprehensive set of perspectives were gathered on the core research themes. Participants stratified by age and gender were enrolled, 24 in Kibra, urban, 23 in Kikuyu, peri-urban and 23 in Nachu, rural setting making a total of 70 participants. Each focus group discussion had a maximum of 6 participants for a total of 12 FGDs. All potential participants were mobilized from the community setting, none of the participants who visited the study venue declined to participate.

### Study area

In 2018, Nairobi had an estimated 25,000 adolescents living with HIV, the highest number nationally. Kiambu ranked fifth among Kenya's 47 counties, with an estimated 5,000 adolescents living with HIV [23]. The study was conducted in Nairobi and Kiambu Counties. These two Counties were selected due to their access, previous working history and HIV burden. In Nairobi, the study activities were conducted in Kibra, while in Kiambu county they took place in Kikuyu Town and Nachu. Kibra provided an urban setting while Kikuyu provided the peri-urban and Nachu the rural settings. The study team worked with community-based organizations in these areas to recruit adolescents aged 19 – 24 years. In all the three study settings, the interviews took place in a church compound hosting adolescent friendly services or programs. There were no non-study participants in the focus groups discussions. The participant demographic information is in **Table 1**. The focus group discussions took place between February 3, and February 6, 2024. A recruitment guide (S1 Appendix) was used to introduce the study to participants.

### Data collection

All participants provided written informed consent prior to taking part in the study procedures (S2 Appendix). A semi-structured FGD guide (S3 Appendix) was used to administer the focus group discussions (FGD). No repeat FGDs were conducted. FGDs were conducted in English and Swahili and audio recorded. The interviewer had a notebook that was

**Table 1. Participant demographic characteristics (n = 70).**

| Variables | | Female (n = 36) n(%) | Male (n = 34) n(%) |
|---|---|---|---|
| **Age (Median)** | | 21.5 (19 - 23) | |
| **Study sites** | Kibra (Urban) | 12 (33.3) | 12 (35.3) |
| | Kikuyu (Peri-urban) | 12 (33.3) | 11 (32.4) |
| | Nachu (Rural) | 12 (33.3) | 11 |
| **Education (Highest Level)** | College | 4 | 4 |
| | Primary | 1 | 5 |
| | Secondary | 31 | 25 |
| **Occupation** | Employed | 2 | 5 |
| | Housewife/ husband | 2 | 0 |
| | None | 14 | 3 |
| | Other specify | 7 | 2 |
| | Self employed | 4 | 5 |
| | Student | 7 | 19 |
| **Access to internet** | Yes | 35 | 34 |
| **Devices you use to access internet** | Computer | 1 | 9 |
| | Laptop | 6 | 15 |
| | Phone | 34 | 33 |
| | Tablet | 0 | 3 |
| **Access internet from a cyber café** | Yes | 35 | 32 |
| **Ever taken a HIV test** | Yes | 34 | 21 |

used to collect any relevant field notes. Each of the FGD took between 45 minutes to 1 hour. Data saturation was not discussed. None of the data transcripts was returned to the adolescent participants for comments or corrections.

## Data analysis

Transcripts were uploaded to MAXQDA (version 24) for data management and organization during the analysis process.
We used the thematic analysis process, starting first with immersion in the data followed by collaborative codebook development. Themes were developed primarily deductively from the focus group discussion guide, with the purpose to generate meaningful findings on knowledge, awareness and information gaps on HIV, mental health, substance use, SRH, nutrition and IPV. A few themes emerged inductively from the data after the first round of data immersion which were also added to the codebook. All three coders then completed a first round of initial coding on one transcription. We then had a meeting to discuss and resolve conflicts, meanwhile also refining the codebook collaboratively to ensure clarity and reliability of coding. Each transcript was coded twice and split up evenly between the three analysts who were coding (MBN, WH, PR). Discrepancies were resolved through regular discussions between the two coders assigned to that document. Memos were used to mark out important issues of note and questions from coders as they arose. The COREQ guidelines were used to guide reporting on this study in the manuscript.

## Reflexivity statement

Two qualitative researchers, a male and a female with a few years working in adolescent related research and interventions in multiple settings, (JS and GM) conducted the FGDs while. Both researchers participated in the consenting

procedures. The data collection team undertook an online human subject protection training and a one-day training on the study procedures and roleplaying.

The three researchers who conducted the coding and analysis were all female with experience in global health based at an American institution (MN, WH, PR). Two of them were studying for a PhD in public health (MN, PR) and the other analyst had completed a bachelor's of medicine and surgery (MBBS) and was completing her master's in public health at the time of data analysis (WH). Two of the researchers were from Asia (WH, PR) and the other from the USA (MN). All had qualitative analysis experience in a global health context from previous research experiences. The principal investigator (PM) held regular meetings with the analysis team to discuss each step of the process and build consensus.

The informed consent document provided information on goals and reasons for doing the research study for participants. Nothing was documented regarding the interviewer's bias, assumptions, reasons and interests in the research topic.

### Ethics statement

Ethical approval was obtained from the Kenyatta National Hospital – University of Nairobi Ethics and Research Committee protocol number P655/09/2023. Written informed consent was also obtained from all the study participants before participation. All participants also consented to the study findings being published in peer reviewed journals.

## Results

A total of 12 focus group discussions were conducted. There was a total of 70 participants. Participants were primarily Secondary school graduates, not employed for female or students for male participants, with access to the internet and mostly accessing the internet via their phones. No participants dropped out from the study or refused to participate.

Major themes derived from the data included: HIV Knowledge and Experiences, Sexual and Reproductive Health, Nutrition and Healthy Lifestyle, Violence and Safety, Substance Use, Mental Health and Emotional Well-being, Digital Health Information Seeking, Barriers to Digital Access, App Design and Functionality, Governance and Safety of the App, Trusted Sources of Health Information. Coding tree provided in Appendix (S4 Appendix and S5 Appendix).

### Youth health

**Sexual health and reproductive health.** In the discussion on sexual and reproductive health, participants touched on different topics like family planning and contraception, sexual health education, menstruation, sexually transmitted infections, and pregnancies:

*"I would like to discuss about family planning because young girls are using family planning and then it later complicates their lives."* (Respondent 3, FGD1)

In addition, participants also mentioned issues like circumcision and its cultural and health implications:

*"On the issue of reproductive health, the parts of the body for the males and females, what we should do with those parts and what we should not do, like the girls are getting circumcised we are also circumcised so we get to know who should be circumcised."* (Respondent 4, FGD12)

**HIV prevention, care and treatment.** HIV was a common sexual health issue which was discussed. Participants highlighted various important issues related to HIV prevention and education. One participant mentioned the need to teach everyone how to use PrEP, condoms, and other protective measures against HIV. While another participant wanted to know what PrEP and PEP look like. There was also a lack of awareness about female condoms, one participant mentioned

*"...I did not know if there is a female condom I was just taught in Dreams, I knew that it is only for men because you can easily find them here in Kibra, even being thrown everywhere even little children know that this is a condom but for the female condom you cannot know if you are not told, I was just told there are female condoms the other day."* (Respondent 4, FGD9)

It was also emphasized to normalize HIV check-ups for both boys and girls to ensure everyone feels comfortable with testing and care. One of them also mentioned the need for better education on the use of self-test kits.

**HIV transmission.** There were major gaps in knowledge regarding HIV transmission highlighting a need for comprehensive education on HIV transmission, one participant mentioned

*"......you know many people do not know how HIV spreads because some people think it spreads through sharing of cups and plates or sneezing or coughing so I think they should put on how it spreads and prevention, how you can prevent HIV."* (Respondent 5, FGD9)

Participants wanted to know how these HIV drugs work and if someone on treatment can still spread the disease. One participant raised a question about whether someone with HIV could transmit the disease by having sex with a girl who is on her period. Another one added to this concern stating whether a man who has sex with an infected woman and then immediately has sex with his wife could pass the virus to her.

**Knowledge and awareness of major health issues**

**Side effects.** One participant discussed the physical and social side effects of living with HIV and focused on the broader impact it has on individuals, their families, and their communities.

*"Talk of the side effects that HIV can bring to you personally in terms of health even if you use those drugs, also how can it affect other people around you, even your families and other people close to you maybe."* (Respondent 5, FGD7)

**Living with HIV/living with someone who has HIV.** Some participants mentioned that they needed support regarding how to live a life while managing HIV. They emphasized the gap of information that was needed regarding medical information, but also in addressing the emotional and practical challenges faced by individuals who live with HIV.

*"I think the major thing that you should discuss is how to live with it, how to live like you, how to, like, just be you with HIV."* (Respondent 1, FGD6)

In discussions around the topic of living with someone who has HIV, they mentioned the need for education on how to avoid getting it while maintaining healthy relationships. They also highlighted misconceptions and stigma in relationships where one partner is HIV-positive, and the other is not. They suggested that clear guidelines and emotional support could help reduce fear and discrimination which would allow individuals to provide love and care to those living with HIV.

*"How can we live positively with these people because some people have the mind that they can't share even a meal with these people if they know them so if today, I know somebody who has this disease I can try as much as possible to unfriend such a person so how can we avoid this? We have the mentality that I can contract HIV anytime."* (Respondent 3, FGD7)

**Menstrual health.** Several important issues surrounding menstruation were raised by participants. Some of these were the accessibility of sanitary towels, menstrual hygiene, and the physical discomfort associated with painful periods.

*"They should teach our females in Kibra how to keep hygiene like when you walk in Kibera you will see pads disposed everywhere."* (Respondent 2, FGD11)

Male participants also shared their perspectives on menstruation, pregnancy, and reproductive health. They showed curiosity and desire to better understand how to support women during their menstrual cycle. Also, there was a clear lack of understanding among young men about how the menstrual cycle and fertility are connected:

*"The way I hear when a girl gets her period, I hear them saying the period comes in a month three times. When you sleep with that girl, can she get pregnant?"* (Respondent 4, FGD3)

**Teen mothers/ teen pregnancy.** There were a range of concerns regarding teen pregnancy and mostly were focused on physical, emotional, and socioeconomic challenges that young mothers face. They discussed the stigma and lasting impact of pregnancy on a girl's future particularly concerning education and economic independence. It was also mentioned that young mothers do not return to school due to embarrassment or sometimes schools do not allow them back.

*"It is not good because [at] the time that you are taking care of the child you could have done something that would develop your life, but you are here taking care of a child and it is not yet your time, you are not stable in life and you already have the burden of a child."* (Respondent 4, FGD3)

One participant also mentioned the need for better education and prenatal care particularly in underserved areas where women might not have access to adequate health information and might engage in risky behaviors

*"Yes, because here in the slum you will find a pregnant woman in the bar drinking, so they should be educated about that part because many are affected."* (Respondent 3, FGD12)

**Sexually transmitted infections.** There were varying levels of awareness about sexually transmitted infections (STIs) among participants. Many expressed a desire for accessible and detailed information regarding awareness, transmission, and treatment options. There were misconceptions and gaps in knowledge, such as confusion between a urinary tract infection and STI and myth about STI transmission through shared toilets.

*"We are aware that these STIs are not only brought by sex, it also depends on cleanliness. So, the adolescents need to be aware on how to do the cleaning."* (Respondent 6, FGD1)

Stigma also seemed to be a big issue surrounding STIs, particularly the fear of disclosing that they might have an infection to a doctor. They emphasized the need for a private, youth-friendly platform where they can discuss and seek advice without the fear of judgement:

*"It is hard to go to hospital to tell the doctor that you have Gonorrhea."* (Respondent 4, FGD11)

**Nutrition.** Most of the participants mentioned nutrition and related concerns focused on food access, balanced diets, and the importance of nutritional education for adolescents especially in communities where poverty limits access to a variety of foods.

*"Many people tend to think that having money is, you will take good meals. They get to understand that even if you are affording cheap meals, they can be cheap, but they are healthy."* (Respondent 2, FGD5)

There was also a mention of personalized nutrition information where an educational tool can take into account the unique circumstances of individuals.

> *"The app should give advice, if you have a certain disease, maybe this is the amount of food you're supposed to take at a certain time."* (Respondent 4, FGD5)

**Substance use.** Substance use was mentioned frequently in the focus groups as an issue which youth are facing. Specific drugs which were mentioned included alcohol, smoking tobacco, bhang, shisha, miraa, and cocaine. Substance use was often mentioned as the source of many other issues, from loss of jobs to gender-based violence, to even death through suicide:

> *"I would recommend most of them who have been using drugs they may have a problem, they may lose their jobs, or they may lose money for the rehabilitation." (Respondent 2, FGD1)*

Participants also mentioned that substance use, especially drinking alcohol, could be linked to violent behavior:

> *"You will speak of alcohol the effect that it will bring let's say the father takes alcohol and becomes violent at home what are the side effects that can affect the children I think children can grow with that and I don't think they live well the way children expected to grow with some comfort I think you will have to emphasize on that."* (Respondent 4, FGD7)

While many participants seemed to know that rehabilitation centers were available for those who wanted help with substance use issues, they also mentioned that these centers were expensive, indicating that these were not services that were accessible to them and their peers.

**Mental health.** Mental health often came up in the discussions about substance use, indicating that the youth thought that the two issues go hand in hand. Youth additionally talked about peer pressure stress as predecessors to mental health issues such as depression and even death or suicide:

> *"…the person can lock himself or herself in a room it can affect and after some time you hear this person has committed suicide, so those are the things you need to make people aware of."* (Respondent 4, FGD7)

One participant also mentioned that their mental health could be affected if they learned they had tested positive for HIV, demonstrating how mental health was also related to their sexual and reproductive health issues:

> *My opinion is, there is no way you are coming to tell an adolescent to abstain whereas, we have been exposed to these things at a very early age. Why can't you tell us ways to deal with your sexual, like, if I do sex, how am I going to protect myself from HIV, from pregnancies, from whatever?* (Respondent 3, FGD6)

One or two participants mentioned that medications could possibly help, but there was not a lot of discussion about how mental issues could be helped in their community.

**Stigma.** Stigma came up a few times, especially when discussing HIV-related issues. Participants who spoke about stigma discussed how ingrained it is in their culture and how they grew up to discriminate against people who have HIV:

> *We've grown up to places that you see a person with HIV, you see a person is slim you think they have HIV, you've grown in a place like, if you see a person being positive and you sit there then you will be positive…* (Respondent 3, FGD6)

They mentioned in relation to a future health app that it is important that the app be inclusive so that people experiencing stigma could feel comfortable joining and finding the support that they need:

*…just ensure that each and every person associated with created support group are actually meant to be there and not any other person who is not supposed to be part of group who is there to intimidated or stigmatized anyone.* (Respondent 5, FGD5)

### Violence

**Intimate partner violence.** Intimate partner violence between the partners in a relationship was one of the areas the youth in this study wanted more information. Some participants mentioned the resources to identify the red flags early in the relationship, resources to report the violence behavior, and support groups to discuss them were needed.

*What I can say about that is, for violence it cannot get to a point you butcher someone like that at one day, it starts with slapping someone once and it escalates from there so for us as women and men we are both precious,* (Respondent 4, FGD9)

Participants wanted support to reduce the anger in triggering situations.

*If you have a temper and you just want to fight all the time how to control that temper, when someone wrongs you what you can do to reduce that temper.* (Respondent 4, FGD11)

### Youth social media usage

### Patterns of social media use among AYP

Though participants varied widely in their reported estimates of their personal and peers' daily social media usage, ranging from 1-24 hours, most reported using it an average of about 7 hours per day. Non-health related reasons for social media usage included entertainment reasons, such as to watch videos, follow sports, read about gossip and celebrities, downloading or listening to songs, for gambling, pornography and cyber-bullying. However, other reasons were more functional in nature, such as watching the news, searching for jobs, working online, studying or simply researching various topics. Some reasons were more passive in nature, where participants mentioned that they would use the internet when they were stressed, that they were addicted to the internet, because it was available, or they had nothing else to do. When asked if they use social media to learn about their health, some participants mentioned that they did not use the internet to learn about their health. As one participant mentioned,

*"I said no because health becomes an issue when you are affected, personally I will not use my social time to search for issues about health, that is why people go to see the doctor when they are sick. That is why I said we as youths will never search for health issues, also we approach the internet on matters concerning health after we have approached our friends and they have said that is not normal so approaching the internet after an influence of the friends, so I will say no."* (Respondent 2, FGD7)

However, those who said they did use the internet, or social media would primarily use it when they or someone they knew had a health problem to learn more about an illness, confirm an illness based on symptoms, or determine what medicine to take. Other reasons included checking on health, to talk to a doctor, verify or better understand health information, seek treatment, search for hospitals, determine the cost of treatment, and look for pharmaceutical drugs.

When discussing reasons for use of social media, how they use it, and the possible consequences of using social media, participants appeared to make statements about how their actions impacted their attitudes and other behaviors. Therefore, as part of this analysis we created a mind map to help visualize how some of these concepts appeared to be linked together (Fig 1).

### Barriers and facilitators to social media and internet use

Barriers to use of social media or internet for health information included "ignorance" and issues in understanding content:

> *"For me it is not easy to Google health, but when I scroll and see health issues, I take time to read and understand, but it is not easy to find me searching." (Respondent 3, FGD11)*

Some mentioned that it was difficult to access because of the cost of data however:

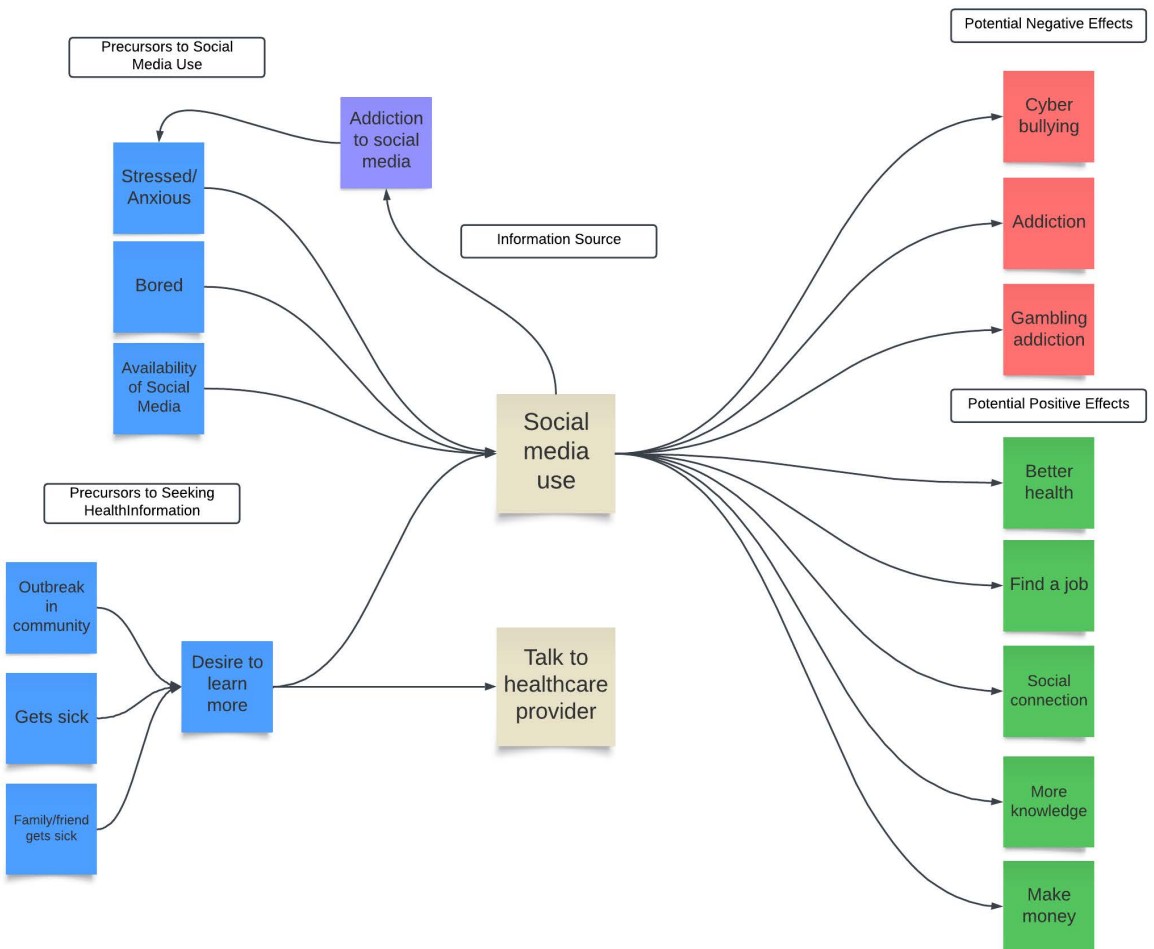

**Fig 1. Mind Map created by our study analysis team to visualize how concepts around social media use were connected based on conversations with AYPs in Nairobi, Kenya (n=70).**

*"According to me I can say that it depends maybe with how much data you have bought to browse or to use on the internet, maybe you can use for one hour." (Respondent 2, FGD9)*

Some participants mentioned that it is more convenient to use the phone to access information on social media however:

*"I will say I use it because it saves time. If I need to know something from the internet, I will use less time rather than going to another institution. I will get the same information faster."* (Respondent 5, FGD8)

A concern some youth had was that sometimes information was not trustworthy if found on the internet:

*"For me anytime I feel unwell there are chemist where the owner is a doctor in Kenyatta so I normally go there because I know the internet is not a doctor and it can also deceive at times. So, the explanation of a doctor is better than the one you get on the internet."* (Respondent 3, FGD12)

### Trusted sources of health information

Most participants mentioned that they would like to see health experts such as healthcare professionals and counselors on the proposed health app which was discussed hypothetically during the focus group discussions. They mentioned that these experts should be present, ready to help, and most importantly be non-judgmental. They mentioned that these experts should listen, guide, give information, answer questions and possibly even refer out to other services if need be. One participant also mentioned that experts should confirm that youth understand the information they are told:

*I know we are not in this question but I would propose something maybe after you have been attended to by the specialist maybe you were undergoing a mental problem maybe you were advised on the things you should do maybe after you are done with the treatment maybe he should provide small ways to see if you have understood the things. (Respondent 3, FGD8)*

Many participants mentioned that they and others in their circles place high trust in medical doctors and other healthcare professionals, but one participant mentioned other sources affiliated with traditional medicine such as witch doctors.

## Recommendations for a future health app

Along with discussing their health experiences and priorities, participants were asked to comment about suggestions they may have for a future hypothetical health application.

### Ground rules of app

Participants were asked to comment about ground rules that they would suggest for the app. Some suggested that there be an age limit for the app, such as over the age of 18, while still others said that there was no need for an age limit since anyone could need this kind of information regardless of age:

*"Even those that are above 24, those that age 27, 28, they have a lot of challenges. If you listen to them, they have a lot, you will be surprised. So it's good we mind about them also."* (Respondent 2, FGD2)

Participants mentioned often that it is important to have good etiquette on the app. They discussed often that there should be standards of types of behavior that was acceptable on the app and some even suggested that participants who did not adhere to the rules should be kicked off the app:

*"There is a way to track people who are intrusive in your business and they are talking badly and there should be a rule or a way of punishing after a few lessons people will cooperate."* (Respondent 5, FGD3)

Finally, another important feature of the app that participants mentioned when speaking of ground rules was privacy and confidentiality. A few participants mentioned that it may be an important rule to have all participants assigned code names to avoid issues of being identified by others when talking about sensitive health topics:

*"I think we should not give names, we should have codes like the ones you gave us"* earlier. (Respondent 2, FGD11)

### Accessibility of a future app

Regarding recommendations that participants gave about a future health app, they emphasized the need for the app to be available at all times, "24/7". When asked about cost, the participants gave a clear response that this app must be free or at least very affordable to ensure people access it. As one participant said,

*"As youths we like free things so when I open this app and I am finding premium subscribe I am gone <<<laughter>>>"* (Respondent 5, FGD8)

They even mentioned that the cost of internet access itself was cost enough that could even cause them to sacrifice meals as a result at times. Participants also mentioned that advertisements should be minimal on the app to ensure they and their fellow youth do not get tired of it and stop using it.

### Contextual tailoring of a future app

Multiple participants emphasized that content should not be only in the written form because youth would not be as interested. They rather suggested that content be presented in images, videos, podcasts, and other engaging ways (see all app recommendations in S5 Appendix).

A few participants also mentioned that they wanted the app to be tailored to individuals and their particular circumstances, recognizing that individuals are not all the same in their needs. For instance, they mentioned that it may be helpful to gender-match participants to experts they discuss certain issues with because they may not be comfortable talking to another gender about that issue.

### Participation and inclusion on a future app

Some participants they mentioned wanting to see on the app included pharmacists, researchers, and even those not affiliated with health such as religious leaders, parents, entertainers, teachers and policymakers. Participants also overwhelmingly wanted to see peers on the app to discuss their issues and learn from each other's experiences. They particularly mentioned this in the context of mental health:

*"I think the app should also try to emphasize on the importance of open conversation among youths because what is the like, what is the importance of a youth talking to maybe an older person or going to an expert when they feel like they are getting depressed or when they are stressed so that they can have at least an open conversation to at least release the stress or what is causing depression in their heads."* (Respondent 4, FGD 5)

## Comparisons between subgroups

### Male and female comparisons

The male and female participants both brought up intimate partner violence, teenage pregnancy and menstruation-related issues. Male and female participants both proposed ideas regarding how the app should be designed, like an idea on how

to provide maps to aid accessibility of health care resources from a male group, and ideas about different ways to present information like podcasts from a female group. However, female participants were more likely to discuss the ground rules on the app in depth, especially the age limit and privacy issues. Female participants were more likely to discuss financial issues than males. Female participants were also more likely to mention the desire for their peers to be on the app than male participants.

Male participants spoke about suicide and death more than females. Male participants also discussed peer pressure more than female participants. Male participants were more likely to discuss issues of accessibility of the information than females while female participants were more likely to discuss reliability and trust of information sources. Male participants also discussed the importance of exercise and first aid being health issues discussed on the app.

## Discussion

### Summary of findings

AYPs in Kenya report using social media frequently, although the reported time of use varies significantly for different individuals. Reasons for using social media include availability of social media apps, addiction, entertainment purposes, use for finding a job or for an online job, and more. In January 2024, it was estimated that there were 13.05 million, 23.5% of the total population active social media users in Kenya [29]. At least 1 in every 5 Kenyans is an active social media user, the majority being adolescents and young people [30].

Participants mentioned that if they were to use a social media app for health purposes, they would likely use it to find information about a disease they or a family member has and how to address that issue, or how to maintain a healthy life. As research shows, a number of social media users rely on it as a source of health information [31]. Some of the social media users connect with healthcare education and information [32].

On the social media app co-designed in our study, the participants mentioned interest in having peers and health experts on an app, but expressed that it is important to maintain ground rules and prioritize confidentiality when having these social media apps meant to improve their health. The healthcare system seems to be embracing use of digital health interventions to offer medical information, patient monitoring and other interactions. However, privacy and security concerns are still a concern [33].

The adolescent users have also suggested important features for a future health app such as an ability to refer them out to other health care providers and services in the real world, reminders, telemedicine options, and multimedia content to engage with. Research has shown that such digital health interventions could be a safe choice to conveniently provide accessible healthcare [34] especially to adolescents and young people who face unique challenges accessing care [35]. Well-designed interventions could leverage artificial intelligence-informed digital health technologies [36].

### Contextualizing in the literature

Our study findings relate to other similar studies among adolescents and young people in the field. One study found that there are low quality sexual and reproductive services and information available for young people in Kenya [37]. Another study found that social media is a popular channel of communication, and a potential source for sexual and reproductive health information [38]. Another study conducted in Ghana, Kenya and Vietnam reported that young adults like to utilize online sources for health information, primarily trusting their peers, similar to our study. This study additionally emphasized the importance of social media health champions, which did not come up as much in the current study [39]. A previous study in Nairobi, Kenya found that compared to the general population, older youth were the most appropriate group in which to utilize DHIs [30]. This further demonstrates the importance of DHIs and their potential to increase engagement among youth. There are demonstrated ways to help guide the development of digital tools to ensure acceptability, intention to utilize and validity in different populations that have been tested in Kenya [40].

## Strengths and limitations

This study builds upon previous work conducted by the principal investigator of this study, finding that youth in Kenya require up to date sexual and reproductive information to guide their decision-making [41]. The current study demonstrates that there is a need for other adjacent health issues such as substance use, nutrition and more, while also providing clear directions for possible DHIs in the future for this population.

This study had a few limitations. Firstly, facilitators of focus group discussions would not always interview in the same way which may have made the data slightly less comprehensive than it might have been otherwise. Secondly, interviewers would switch between talking about social media and the internet as if they were the same thing which led to minor inconsistencies in the data collected. Lastly, some facilitators may have asked leading questions which may have brought up issues which participants may not have brought up organically. This emphasizes the need for rigorous training, follow up and feedback mechanisms, and debriefs where facilitators discuss. Additionally, the questionnaire could have been a bit structured in its approach, leaving little room for new themes to emerge from the participants. Future research could consider taking a more open-ended approach to guide development of frameworks or theories to inform future interventions.

## Future directions

As mentioned in the results above, privacy and confidentiality are important issues for AYPs and should be considered carefully in the design and implementation of future digital interventions. However, interactions with both peers and professionals are important for AYPs. They need peers for this age group to discuss these issues and not feel alone but also need the professionals in order to be able to get the most accurate and trusted information and avoid misinformation.

Accessibility of digital health apps should be prioritized. For digital interventions, in particular, cost of data and use of the apps are important barriers to overcome to ensure youth use the app. To address this barrier, it may be possible to work with cell phone companies to allow all charges of the app to be covered by an NGO or clinic rather than by the participating client. If these recommendations are implemented, youth in Nairobi may consider changing their social media habits and start using their phones to help improve their health. Social media and phone apps have the potential to create a large impact among AYPs and improve outcomes such as mental health, substance use, health misinformation, and sexual and reproductive health.

There is a need to explore the utility of artificial intelligence (AI)-based digital health interventions especially in resource constrained settings like Kenya where healthcare procession shortage limits access to care. The AI-based interventions need to be contextualized, aligned and validated to the local settings. A user-centered approach to the design, development and prototyping of adolescent-targeted digital health interventions shows great promise and potential impact.

## Conclusions

DHIs have great potential for increasing health-related knowledge among youth in Kenya in a number of different areas, including sexual and reproductive health, substance use and nutrition. A user-centered design approach may have great promise to increase cultural sensitivity and ensure acceptability and usability in this population [42,43]. Ideally, this process should be led by the youth themselves, or at least peer leaders [44].

## Supporting information

**S1 Appendix. Focus Group Discussion Recruitment Script.**
(DOCX)

**S2 Appendix. Focus Group Discussion Guide.**
(DOCX)

**S3 Appendix. Informed Consent Form Focus Group Discussion.**
(DOCX)

**S4 Appendix. Coding Tree for the Deskes qualitative analysis studying health issues among AYPs in Kenya and their recommendations for a health. app**
(DOCX)

**S5 Appendix. Recommended features of a future health app for youth in Nairobi Kenya.**
(DOCX)

## Author contributions

**Conceptualization:** Paul Macharia, Cyrus Mugo, Christine Ngaruiya.

**Data curation:** Paul Macharia, Priyanka Ravi, Wadana Hamzazai.

**Formal analysis:** Maiya G. Block Ngaybe, Priyanka Ravi, Wadana Hamzazai.

**Funding acquisition:** Paul Macharia.

**Investigation:** Paul Macharia.

**Methodology:** Paul Macharia.

**Project administration:** Paul Macharia, Maiya G. Block Ngaybe.

**Resources:** Paul Macharia.

**Software:** Paul Macharia, Maiya G. Block Ngaybe.

**Supervision:** Paul Macharia.

**Validation:** Paul Macharia.

**Visualization:** Paul Macharia, Maiya G. Block Ngaybe.

**Writing – original draft:** Paul Macharia, Maiya G. Block Ngaybe, Wadana Hamzazai.

**Writing – review & editing:** Paul Macharia, Maiya G. Block Ngaybe, Hellen Moraa, Boyani Moikobu, Cyrus Mugo, Christine Ngaruiya.

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
