## [Decision Letter · Decision Letter 0]

16 Dec 2025

PGPH-D-25-03489

Perspectives of adolescents and young people on Digital Health Interventions and their impact on health knowledge

Dear Dr. Block Ngaybe,

Thank you for submitting your manuscript to PLOS Global Public Health. After careful consideration, we feel that it has merit but does not fully meet PLOS Global Public Health’s publication criteria as it currently stands. Therefore, we invite you to submit a revised version of the manuscript that addresses the points raised during the review process.

We look forward to receiving your revised manuscript.

Kind regards,

Joel Msafiri Francis, MD, MS, PhD

Academic Editor

Journal Requirements:

i. State the initials, alongside each funding source, of each author to receive each grant. For example: "This work was supported by the National Institutes of Health (####### to AM; ###### to CJ) and the National Science Foundation (###### to AM)."

ii. State what role the funders took in the study. If the funders had no role in your study, please state: “The funders had no role in study design, data collection and analysis, decision to publish, or preparation of the manuscript.”

3. Please ensure that your Ethics Statement is available in its entirety at the beginning of your Methods section, under a subheading 'Ethics Statement'.

4. Please upload separate figure files in .tif or .eps format. Also, remove the figures from your manuscript file but keep the legends.

5. We notice that your supplementary information ‘Appendix 1, 2, 3, 4’ are included in the manuscript file. Please remove them and upload them with the file type 'Supporting Information'. Please ensure that each Supporting Information file has a legend listed in the manuscript after the references list.

6. We have noticed that you have uploaded Supporting Information files, but you have not included a list of legends. Please add a full list of legends for your Supporting Information files after the references list.

7. In the online submission form, you indicated that “A deidentified data set may be made available upon request from author PM. Due to the sensitive nature of the topics of conversation, it is better to keep the dataset private to protect the identities of the participants.”.

3. Uploaded as supplementary information.

Additional Editor Comments (if provided):

Reviewers' comments:

Reviewer's Responses to Questions

**Comments to the Author**

1. Does this manuscript meet PLOS Global Public Health’s publication criteria? Is the manuscript technically sound, and do the data support the conclusions? The manuscript must describe methodologically and ethically rigorous research with conclusions that are appropriately drawn based on the data presented.? Is the manuscript technically sound, and do the data support the conclusions? The manuscript must describe methodologically and ethically rigorous research with conclusions that are appropriately drawn based on the data presented.

Reviewer #1: Partly

Reviewer #2: Yes

2. Has the statistical analysis been performed appropriately and rigorously?

Reviewer #1: Yes

Reviewer #2: Yes

3. Have the authors made all data underlying the findings in their manuscript fully available (please refer to the Data Availability Statement at the start of the manuscript PDF file)?

The PLOS Data policy requires authors to make all data underlying the findings described in their manuscript fully available without restriction, with rare exception. The data should be provided as part of the manuscript or its supporting information, or deposited to a public repository. For example, in addition to summary statistics, the data points behind means, medians and variance measures should be available. If there are restrictions on publicly sharing data—e.g. participant privacy or use of data from a third party—those must be specified.requires authors to make all data underlying the findings described in their manuscript fully available without restriction, with rare exception. The data should be provided as part of the manuscript or its supporting information, or deposited to a public repository. For example, in addition to summary statistics, the data points behind means, medians and variance measures should be available. If there are restrictions on publicly sharing data—e.g. participant privacy or use of data from a third party—those must be specified.

Reviewer #1: No

Reviewer #2: No

4. Is the manuscript presented in an intelligible fashion and written in standard English?

Reviewer #1: Yes

Reviewer #2: Yes

Reviewer #1: General comment

It is an important study which addresses emerging issue of digital health interventions among adolescents and young people and their impact on health knowledge

Abstract:

Methods

What type of qualitative study was this (Narrative, phenomenological or …)?

Can you justify use of focus group discussion to study the sensitive topic among adolescent and young people?

Results

Clarify the themes that you found in your study. I find number of ideas explained not clear if these are the found themes or otherwise.

Introduction

You have done great job reviewing literature from globally to East Africa on second paragraph. What about data from study country (Kenya)?

Refer to original document with the respective information for example reference 1, refer to “UNAIDS. The Path That Ends AIDS. 2023 UNAIDS Global AIDS Update. Available at: https://thepath.unaids.org/wpcontent/themes/unaids2023/assets/files/2023_report.pdf. [Accessed 27 April 2024]” instead of “Rakhmanina N, Foster C, Agwu A. Adolescent and young adults with HIV and unsuppressed viral load: where do we go from here?”

Methods

Study area

What is age range for adolescents? Then this statement “The study team worked with community-based organizations in these areas to recruit adolescents aged 19 – 24 years.”

Data collection

At what time was English and Swahili used? Since you used both languages

Why was data saturation not discussed?

Results

The authors should revise this sentence for clarity. It is not clear whether females were predominantly unemployed, and males were predominantly students, or if another interpretation is intended. Additionally, several characteristics (education, employment, internet access) are combined in one long sentence. I recommend restructuring into multiple clearer statements.

In table 1 row 2, it states Age (Median), what is the median age and the number in brackets (19-23)? What is the unit that measured age? Is it years or months?

One suggestion: Age (Years), Median (IQR)

Discussion

Revise the discussion to include all the themes, compare, contrast the findings, and include possible explanations.

Reviewer #2: This is a well-structured and well-written manuscript presenting timely findings that contribute meaningfully to improving access to evidence-based sexual and reproductive health (SRH) information through digital health interventions. The topic is highly relevant, particularly in the current era characterized by widespread misinformation and information overload.

I have a few specific suggestions to strengthen the manuscript:

1. Adolescent Involvement in the Participatory Approach

While the paper notes that a participatory approach was applied, it only briefly mentions the level of adolescent involvement beyond their participation as study subjects. I recommend adding a more detailed description of how adolescents were engaged throughout the research process. This could include clarification on whether they were involved in co-designing data collection tools etc.

2. Gender-Specific Discussion

The study commendably highlights differences in perspectives between male and female participants; however, this component is not sufficiently addressed in the discussion section. Given well-documented gender differences in health-seeking behaviors and digital engagement patterns, it would strengthen the paper to discuss how these findings might inform the design of gender-responsive features within the proposed digital health intervention (DHI).

**Do you want your identity to be public for this peer review?** For information about this choice, including consent withdrawal, please see our Privacy Policy..

Reviewer #1: No

Reviewer #2: No

---

## [Editor Report · Decision Letter 1]

2 Mar 2026

Perspectives of adolescents and young people on Digital Health Interventions and their impact on health knowledge

PGPH-D-25-03489R1

Dear Mrs. Block Ngaybe,

We are pleased to inform you that your manuscript 'Perspectives of adolescents and young people on Digital Health Interventions and their impact on health knowledge' has been provisionally accepted for publication in PLOS Global Public Health.

Best regards,

Joel Msafiri Francis, MD, MS, PhD

Academic Editor